# Effect of Revolutionary Pitch on Interface Microstructure and Mechanical Behavior of Friction Stir Lap Welds of AA6082-T6 to Galvanized DP800

**Shuhan Li** [1,2], **Yuhua Chen** [2,*], **Jidong Kang** [1,*], **Babak Shalchi Amirkhiz** [1] and **Francois Nadeau** [3]

1   CanmetMATERIALS, Natural Resources Canada, 183 Longwood Road South,
    Hamilton, ON L8P 0A5, Canada; shuhanli@outlook.com (S.L.); babak.shalchi_amirkhiz@canada.ca (B.S.A.)
2   School of Aerospace Manufacturing Engineering, Nanchang Hangkong University, 696 Fenghe Road South,
    Nanchang 330063, China
3   National Research Council of Canada (NRC), Saguenay, QC G7H 8C3, Canada;
    Francois.Nadeau@cnrc-nrc.gc.ca
*   Correspondence: ch.yu.hu@163.com (Y.C.); jidong.kang@canada.ca (J.K.); Tel.: +86-133-3006-7995 (Y.C.);
    +1-(905)-645-0820 (J.K.)

**Abstract:** Friction stir lap welding of 1.5-mm thick 6082-T6 aluminum alloy to 2-mm thick galvanized DP800 steel (Zn-coated) was carried out. Optimal welding conditions were obtained aiming to defect-free joints with good mechanical properties. The interfacial intermetallic compounds (IMCs) at the stir zone and hook zone were characterized under different revolutionary pitches. With a revolutionary pitch of 1.0 mm/rev, maximum joint strength reached 71% of that of the aluminum alloy. In the meantime, the average thickness of IMC layer is less than 1 μm; $Al_{3.2}Fe$ in the Al-rich side and $Al_5Fe_2$ in the Fe-rich side at the interfaces of stir zone while $Al_6Fe$ and nanocrystalline close to $Al_{3.2}Fe$ at the interface of the hook zone. At a relatively lower revolutionary pitch (0.5 mm/rev), Zn was found with the aggregation of Si and Mn at the hook-zone interface, leading to the generation of Al-Fe-Si phase thus decreasing the thickness of the IMC layer. In the stir zone, the revolutionary pitch has a significant influence on the interfacial microstructures. The interfacial IMC layer at 1.0 mm/rev is simple and flat, but the one at 0.5 mm/rev becomes thicker and more complex. Stir zone aluminum under different revolutionary pitches is similar in microhardness and tensile behavior. The mechanical response of joints was modeled based on linear mixture law with an iso-strain assumption and neglection of the IMC layer. The modeling results are in good agreement with the experimental ones indicating the resultant interfaces act as good as the good boundaries between dissimilar Al/steel joints.

**Keywords:** Al/steel dissimilar materials; friction stir welding; interface; intermetallic compounds

## 1. Introduction

With the increasing use of aluminum alloy for lightweighting (replacing steel) in the automotive industry, dissimilar aluminum to steel joining has received substantial attention [1]. However, the differences in thermal properties (i.e., melting point, heat capacity, thermal expansion, and thermal conductivity) and formation of brittle Al/Fe intermetallics (i.e., $AlFe_3$, $AlFe$, $Al_2Fe$, $Al_3Fe$, $Al_5Fe_2$, and $Al_6Fe$), welding distortion, cavities, and cracks cause fusion welding of aluminum to steel to be challenging [2].

As a solution, the use of friction stir welding (FSW) [3] helps in reducing intermetallic compounds (IMCs) in Al/steel joints. For example, the subframe assembly of the Honda Accord 2013 lap joined

aluminum to steel using the FSW technique [4]. Bozzi et al. [5] made friction stir spot welds of AA6016/IF-steel. $Al_3Fe$, $Al_5Fe_2$, and $Al_2Fe$ were identified within an optimal IMC layer with a thickness of 8 µm. Uzun et al. [6] friction stir welded AA6013-T4 to 304 stainless steel successfully and the joints could reach approximately 70% base aluminum alloy with an interfacial layer less than 1 µm thick.

However, there are sparse data in the open literature on the FSW of aluminum alloy to advanced high strength steels (AHSS [7]), which are more desirable for automotive applications [8]. Liu et al. [8] friction stir welded AA6061-T6511 (1.5 mm) to TRIP 780/800 sheets of steel (1.4 mm) in butt joint configuration. It was reported that the maximum ultimate tensile strength (UTS) could reach 85% of the base aluminum alloy and the interfacial IMCs layer of AlFe and $AlFe_3$ has a thickness less than 1 µm. Zhao et al. [9] investigated the tool geometry effects on AA6061/TRIP 780/800 (1.5 mm thick) FSW joints. They demonstrated the tool geometry determines the thickness of the IMC layer. With an appropriate tool geometry, the UTS of the joint could be higher than 80% of the base Al alloy. Further, an electrically-assisted FSW system was developed for butt joining AA6061/TRIP 780 by Liu et al. [10].

It should be highlighted that researchers mostly focused on the investigation on butt configuration, while insufficient work has done with friction stir lap welding of Al/AHSS [8–10]. Compared to the butt configuration, the advantage of a lap joint is that it provides no lateral gap. As no filler wire is used in friction stir welding, approximately 20% of the thickness in terms of maximum gap allowed is typically required [11]. It also depends on the underfill value after welding to pass the requirements. Also, the positioning in lap joints is more robust than in butt joints where lap joining does not need to follow the joint line. It can be shifted off slightly without quality compromise and simplify part positioning in production.

In the present study, AA6082-T6 and Zn-coated DP800 were friction stir lap welded for automotive application. Various welding parameters were tested aiming for a process window with optimal welding conditions. The resultant interfacial microstructures were then characterized using transmission electron microscopy (TEM) in addition to optical and scanning electron microscopy (OM and SEM) to reveal the effect of welding parameters on those results. Further, the role of interface microstructure on mechanical properties of the FSWs was demonstrated through experiments and modeling of a series of modified shear tests in addition to lap shear tests.

## 2. Experimental Procedures

The materials used in this work were 1.5-mm thick 6082-T6 aluminum alloy and 2-mm thick galvanized DP800 dual-phase steel. These grades come from Neuman Aluminium through Europe. The surface of DP800 was coated with a 19-µm thick zinc layer. The nominal chemical compositions of the base metals are listed in Table 1. Welding process was carried out using a gantry friction stir welding machine where the pin-tip of the tool slightly penetrated into the steel sheet [12,13] (see welding setup illustrated in Figure 1). As the sub-steel is a high strength steel, it may cause significant wear to the tool pin-tip. To avoid this, the FSW tool material was selected as WC-20%Co with a cylindrical geometry and rounded pin end. Travel speeds were 300, 750, and 1250 mm/min, and rotation speeds were 500, 750, 1000, 1250, and 1500 RPM. Process parameters are proprietary to the NRC research consortium ALTec. All welded joints were prepared using an MTS I-Stir PDS gantry-type FSW machine and specimens were extracted in the steady-state region of the welds, away from plunge and exit zones. The welding parameters were varied in a wide range aiming for a process window with optimal welding conditions. In addition, the longitudinal force during welding was recorded for the potential application of robotic welding.

**Table 1.** Nominal chemical compositions of base materials (wt. %).

| AA6082 | Mg | Si | Mn | Cu | Al | Fe | Others |
|---|---|---|---|---|---|---|---|
| | 1.10 | 0.96 | 0.41 | 0.02 | 96.99 | 0.43 | Bal. |
| DP800 | C | Si | Mn | Ni | Cr | Fe | Others |
| | 0.15 | 0.22 | 1.79 | 0.03 | 0.41 | 97.10 | Bal. |

**Figure 1.** Schematic illustration of the welding setup and details of the tool and materials.

After welding, some of the FSWs were cross-sectioned following the standard metallographic procedure. Microstructural observations were performed via an Olympus OM and a FEI Nova Nano 650 SEM equipped with energy dispersive X-ray spectrometry (EDS). Furthermore, to characterize the interfacial IMC, focused ion beam (FIB) milling was used to prepare the interface specimens using the FEI Nova Nano 650 SEM. Then the specimens were characterized via a FEI Tecnai Osiris TEM for investigation. Selected area diffraction (SAD) was used for phase identification. Scanning transmission electron microscopy (STEM) mode using the bright field (BF) and high angle annular dark field (HAADF) detectors were performed in combination with EDS. Vickers hardness along the cross-centerline of Al sheet was tested every 0.5 mm using a load of 100 g with a dwell time of 10 s. Lap shear specimens were cut in a nominal width of 25 mm and tested using an MTS test frame.

For the mechanical behavior evaluation of joints' stir zone, a modified shear test specimen developed by Kang et al. [14] was adopted to obtain the mechanical response of different stack-ups. The modified shear test specimen with the shear zone central at the weld was designed as shown in Figure 2 (volume fraction of steel is 15%). A commercially available optical strain mapping system based on digital image correlation (DIC), Aramis, was used to follow the strain development during the shear test. Details of this method are given in past papers [14,15]. After the testing, to measure the actual thickness aluminum and steel in the shear zone thus calculate the actual volume fraction of each material, the shear test specimens were cut cross the shear zone followed a standard metallographic procedure for optical microscopy. In addition, mini-tensile specimens of stir-zone Al/steel were designed as shown in Figure 3. All mechanical tests were carried out at a crosshead speed of 3 mm/min at room temperature. Note that the number of the replicate mechanical test specimens is three. Wire electrical discharge machining (WEDM) and sink EDM were used to machine these samples.

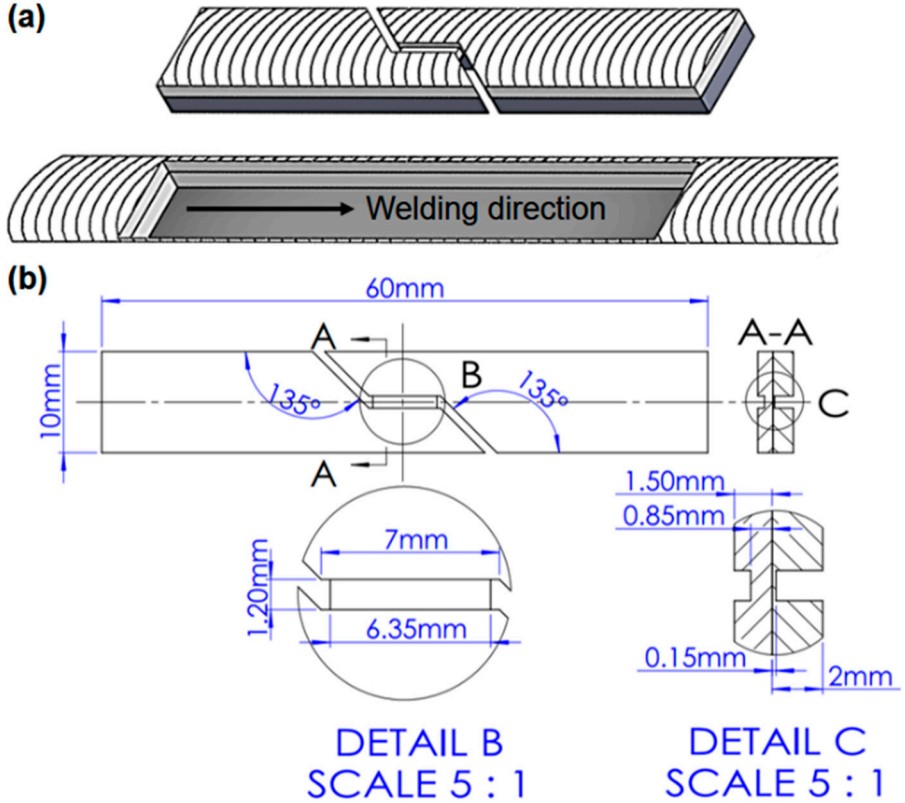

**Figure 2.** A shear test specimen with a volume fraction of 15% steel and 85% Al (**a**) schematic illustration of sampling location and (**b**) specimen geometry.

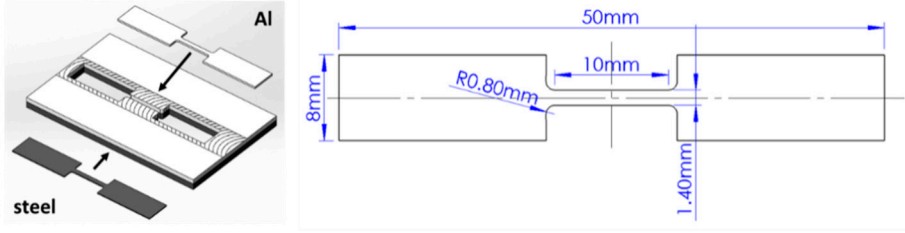

**Figure 3.** Mini-tensile specimens for measuring stir zone Al mechanical responses.

## 3. Results and Discussion

### 3.1. Welding Process Optimization

In general, process optimization mainly targets three objectives: producing defect-free welds, maximizing joining efficiency, and minimizing welding time [16]. In the present study, for a potential robotic welding application, the longitudinal tool force under various welding conditions was also taken into account.

Figure 4 shows the combined effect of rotation speed and travel speed on the joint formation. The points in black rhombuses indicate the good welds without visible macro defects by visual inspection and optical microscopy. The points in open circle indicate the joints with volumetric defects and the solid circle in gray color indicates the joints needing high longitudinal force for welding. The results shown in Figure 4 indicated that the joint formation is quite dependent on the rotation speed but independent on the travel speed. The defects mainly occurred when the rotation speed is less than 800 RPM. Sound joints were obtained with a wide range of welding parameters: rotation speeds ranging from 1000 to 1500 RPM and travel speeds ranging from 300 to 1250 mm/min.

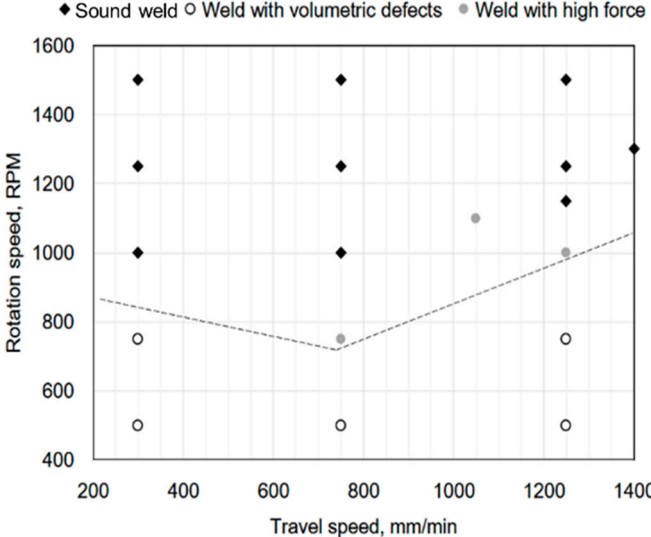

**Figure 4.** Process window of joint formation.

In the following discussion, we eliminate those data points with defects (points in open circles in Figure 4). For the robotic solution of friction stir welding, the effect of the revolutionary pitch (RP) on the longitudinal force is shown in Figure 5. The major issue with robotic FSW process is the high loads induced by the process itself. In our study, the targeted robotic FSW equipment needs a payload that is estimated to be less than 300 kg (3 kN). Thus, an approximate admissible load of 2.25 kN, i.e., 75% of the payload limit, is chosen as the guideline. The joints with RP no more than 0.3 mm/rev were produced under a low force rate, suggesting aluminum was highly plasticized with high heat input during welding. RP ranging from 0.4 to 1.1 mm/rev resulted in a steady rate of the longitudinal force, which almost reached the threshold of the longitudinal force in robotic FSW. It is noteworthy that all three data points in the solid gray circle in Figure 5 were at a lower level of rotation speeds (750–1100 RPM) compared with others joints at RP values approximately 1 mm/rev. This also means that, with the same RP, decreasing rotation speed may significantly increase the longitudinal force during welding processing as the rotational speed has a strong influence on the specific energy which is related to the temperature distribution within a workpiece generated during FSW [17]. As process temperature decreases, the longitudinal forces increase accordingly.

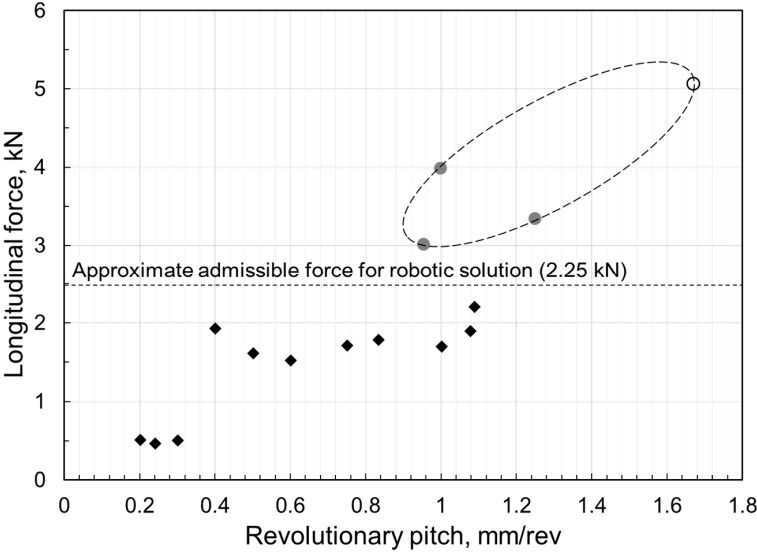

**Figure 5.** Longitudinal force versus revolutionary pitch.

Figure 6 shows the relationship between RP and joints' lap shear performance. The maximum lap shear strength was obtained at RP of 1.0 mm/rev, which is 71% of the respective weaker base material (AA6082-T6). Joints with good lap shear performances (more than 360 N/mm) were obtained at revolutionary pitches from 0.6 to 1.0 mm/rev except using rotation speeds below 1200 RPM. As seen in Figure 5, the longitudinal force at 0.5 mm/rev is at a similar rate compared with those at 0.6 to 1.0 mm/rev.

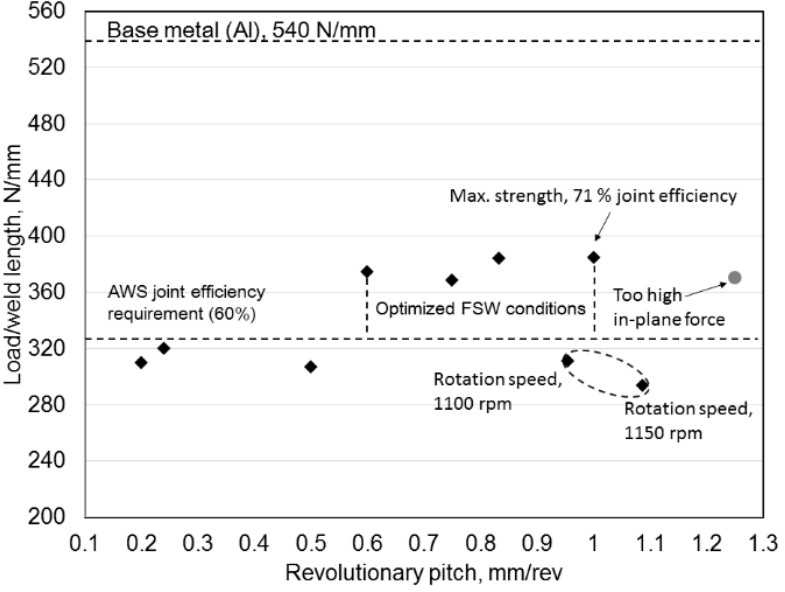

**Figure 6.** Lap shear test results in a function of revolutionary pitch (RP) value.

Considering joint appearance, tool payload, and maximum joint load capacity, 0.6 to 1.0 mm/rev were selected as the optimal revolutionary pitch range in this work. Thus, to understand the welding condition effect on interfacial microstructures and mechanical properties of the FSW Al/steel joints, further investigation was conducted focusing on the optimized joint with the highest RP (1.0 mm/rev). For comparison, the one with less maximum joint load capacity (0.5 mm/rev) was also selected. The detailed welding parameters are shown in Table 2.

**Table 2.** Detailed welding parameters for further investigation.

| Sample | Rotation Speed, RPM | Travel Speed, mm/min | Revolutionary Pitch, mm/rev |
|--------|--------------------|--------------------|----------------------------|
| A | 1250 | 1250 | 1.0 |
| B | 1500 | 750 | 0.5 |

*3.2. Interfacial Microstructure*

Figure 7 illustrates the macrostructures of cross weld section of the two FSWs. When looking at the steel hook on the retreating side, the widths of the thermomechanically affected zone (TMAZ) and SZ of both specimens are different, indicating the potential different interfacial characteristics with different welding heat input.

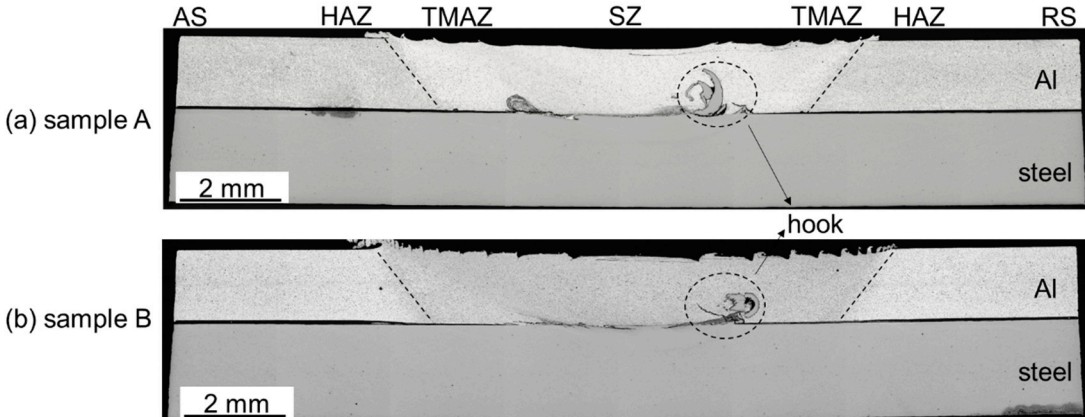

**Figure 7.** Cross-sectional macrostructures of (**a**) specimen A and (**b**) specimen B. (AS: advancing side, RS: retreating side).

Another feature seen in Figure 7 is the black crack-like layer in the hook zone. It was reported that the hook geometry significantly influenced the joint strength [18]. In addition, the interfacial brittle intermetallic compound plays a key role in the joints' interface strength. Borrisutthekul R et al. [19] proposed that, in order to get satisfactory joints, the thickness of the reaction layer should be limited to several micrometers. Therefore, a detailed SEM investigation was performed to obtain microstructures as well as the thickness of the interfacial IMCs layers in different locations (SZ and hook-zone interfaces). The results are shown in Figure 8. As seen in Figure 8a,b, at an RP of 1.0 mm/rev, the mean thickness of IMC layer is less than 1 μm, and the layer at the hook zone of the retreating side is slightly thicker than the one in the stir zone. Figure 8c presents the center zone interface of 0.5 mm/rev. In comparison with Figure 8a, this result indicates that higher revolutionary pitch results in a thinner IMCs layer which is in agreement with previous findings. However, the mean thickness of the interface at the hook zone of 0.5 mm/rev (see Figure 8d) is 0.97 μm which is thinner than the center zone interface at the same welding condition. More interesting is that it is approximately equal to the interface of the hook zone at the higher revolutionary pitch (1.0 mm/rev).

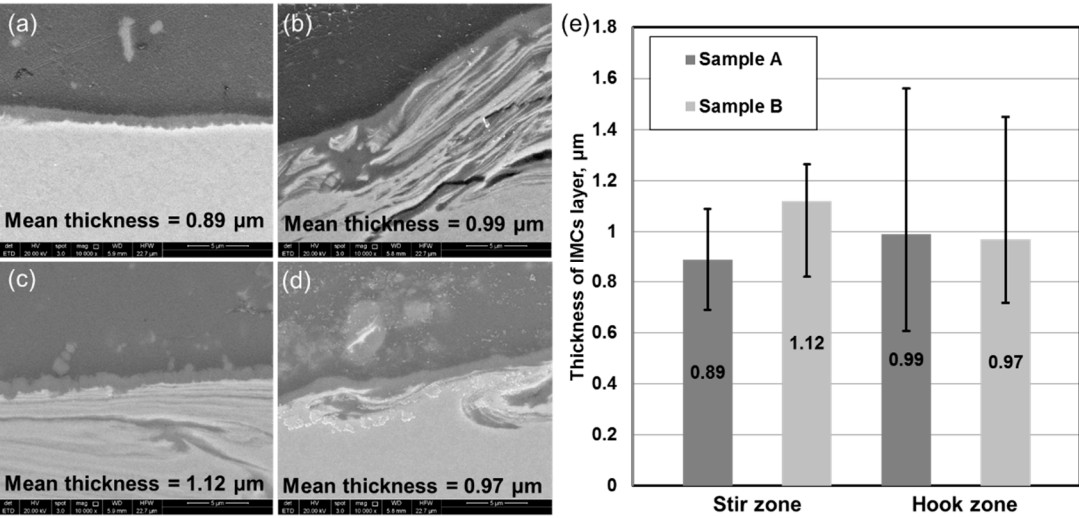

**Figure 8.** SEM images for evaluating the thickness of intermetallic compounds: (**a**) stir zone of sample A, (**b**) hook zone at retreating side of sample A, (**c**) stir zone of sample B, (**d**) hook zone at retreating side of sample B, and (**e**) thicknesses of interfacial layers.

The variation of the interfacial IMCs thickness at hook zone is unreasonable in general understanding. It was reported that zinc has a significant effect on the performance of Al/steel

joints, but this effect may be positive or negative [20–23], suggesting Zn plays a possible influence on interfacial IMCs. In this study, during the welding process, the pin tip was slightly inserted into the steel sheet with a zinc coating layer. In the stir zone with drastic deformation, the zinc coating was likely to be stirred into the aluminum matrix. That means the center interface hardly contain zinc layer. To confirm this, EDS element distribution maps were carried out to trace the zinc element (Figure 9). The results indicate there is no aggregation of zinc in the interfacial zone. The presence of zinc was further checked by TEM investigation. EDS element distribution maps of the hook zone of sample A are present in Figure 10. As seen from the results, the black crack-like layer in Figure 7 is distinguished as the zinc coating. That is to say, at the hook zone, the zinc coating was lifting up and away from the steel surface. The nearly intact morphology of the zinc layer indicates there was no significant dissolution of zinc at the interfacial zone prior to the lift-up of the zinc layer.

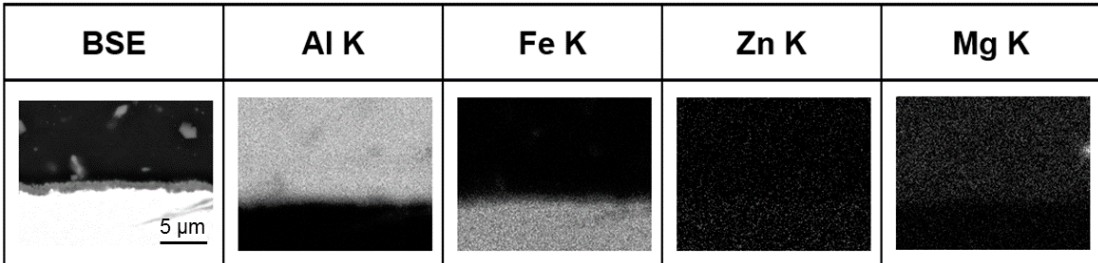

**Figure 9.** EDS element mapping of the center interface of sample A.

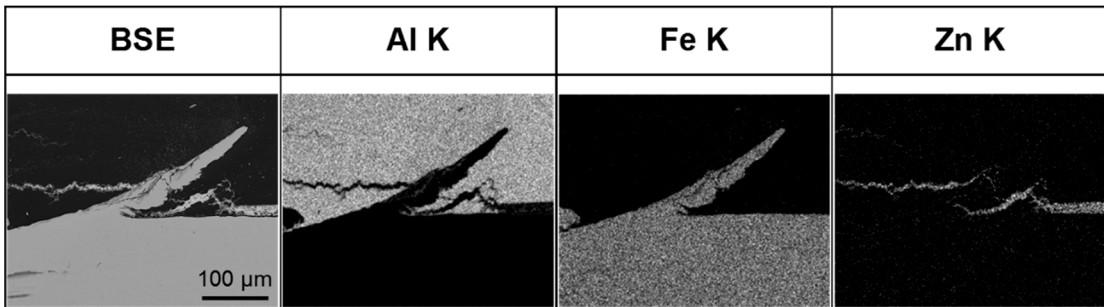

**Figure 10.** EDS element mapping of the hook zone at the retreating side of sample A.

However, the situation is different for the case with the lower revolutionary pitch (0.5 mm/rev). It should be noted that the abnormal thickness of IMCs layer in Figure 8d was observed in the region I in Figure 11a. To clarify whether this is a real reflection of the specimen, the hook zone structures and EDS element distribution maps in different regions were also investigated. Similar to Figure 10, zinc layer rising was also observed in Figure 11a. Figure 11b,c presents the EDS element distribution maps of regions I and II. As seen from Figure 11b, the zinc coating is highly raised up in region I and there is no zinc left. However, as shown in Figure 11c, the rising zinc coating in region II shows signs of dissolution in the aluminum matrix. During the welding process, in comparison with region II, region I is relatively closer to the rotating pin that is easier to be affected by the mechanical effect of the pin. Therefore, in the region I, the rapid rise of the zinc layer causes no dissolution at the interface because the mechanical effect is stronger than the heat effect. In contrast, in region II, due to the limited lifting of zinc, the dissolution of the zinc layer occurred at the interfacial zone. Since no zinc at the interfacial in the region I as seen in the SEM analysis, the abnormal IMCs layer in this region is yet to explain. Thus, region I was further analyzed using TEM and compared with other locations.

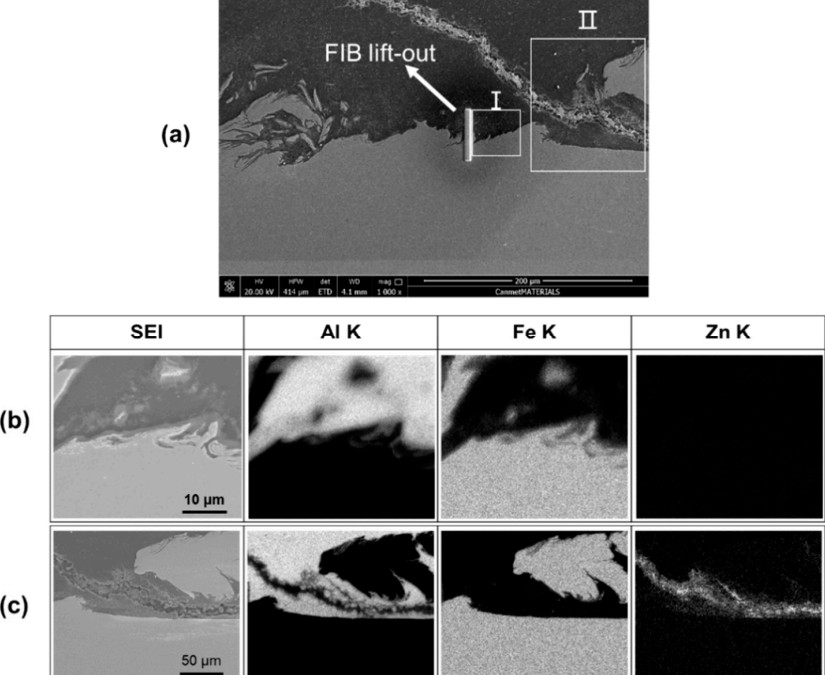

**Figure 11.** Hook zone at the retreating side of sample B: (**a**) SEM image of hook zone and energy dispersive X-ray spectrometry (EDS) element mapping of (**b**) region I and (**c**) region II.

Figure 12 shows the TEM images of the interface in SZ of specimen A. The interfacial IMCs layer reveals a simple flat microstructure (see Figure 12a), which thanks to the high travel speed. The HAADF image of the interface is given in Figure 12b and the relevant EDS combination map is presented in Figure 12c. While the elemental distribution shows no zinc at the interface the thickness of the IMCs layer is $0.5 \pm 0.1$ µm, which is slightly thinner than the SEM results. In addition, two kinds of IMCs were detected by SAD patterns: $Al_{3.2}Fe$ in the Al-rich side (position d in Figure 12b) and $Al_5Fe_2$ (position e in Figure 12b) in the Fe-rich side. The distribution of interfacial IMCs is consistent with the reported results of Al/steel friction stir welded joints [12,24].

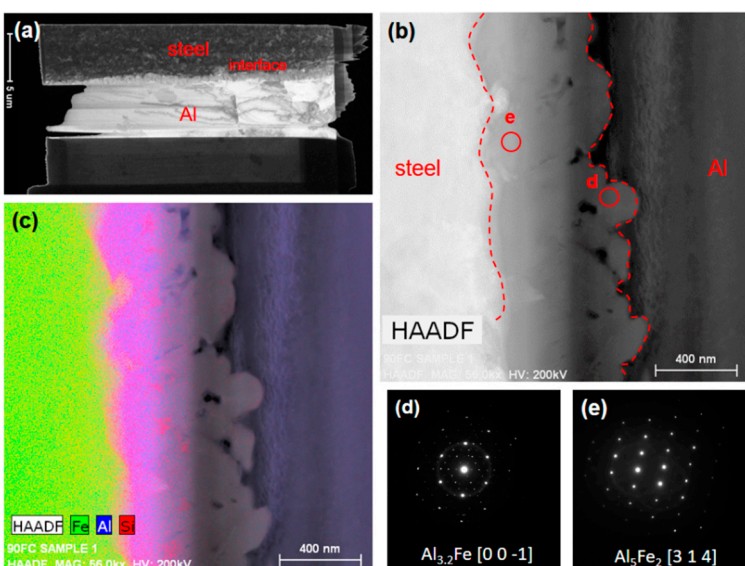

**Figure 12.** TEM micrographs of the interfacial microstructure at SZ of sample A: (**a**) whole view, (**b**) HAADF interfacial image, (**c**) EDS combination elemental map and SAD patterns of (**d**) Position d and (**e**) Position e in (**b**).

The TEM images of the SZ interfacial layer in specimen B are given in Figure 13. Similar to specimen A (Figure 12), $Al_{3.2}Fe$ and $Al_5Fe_2$ were also identified at the interface at the same order. In comparison with Figure 12a, a significant difference as seen in Figure 13a is the needle-shaped objects scattered in the aluminum matrix (higher magnification view in Figure 13b,c), which were identified as $Al_{3.2}Fe$. The interfacial reaction layer is thicker than that in specimen A which also agrees with the SEM analysis (Figure 8). Note that no zinc was found in both elemental distribution maps as shown in Figures 12c and 13c. That is to say, in the stir zone, with the pin inserted into the steel, the zinc coating was apart from the original location and no zinc remained at the interface.

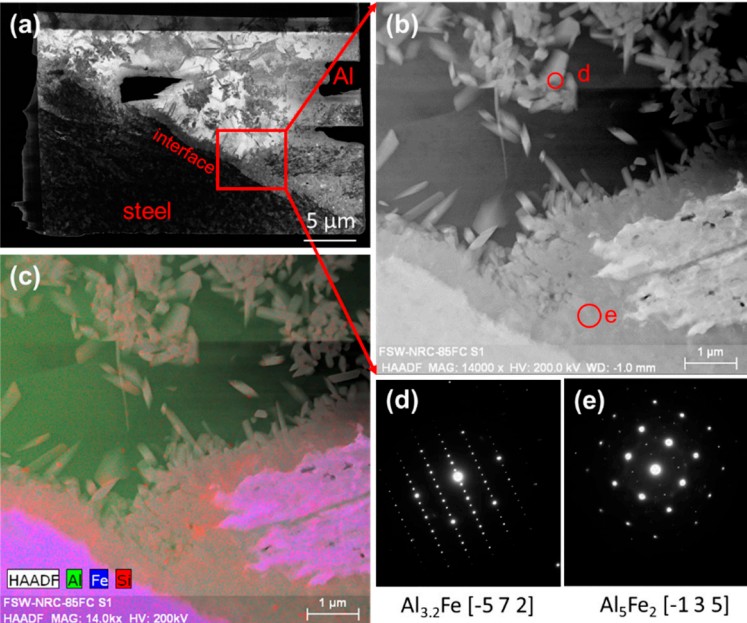

**Figure 13.** TEM micrographs of the interfacial microstructure at the stir zone of sample B: (**a**) low magnification STEM images, (**b**) high angle annular dark field (HAADF) interfacial image, (**c**) EDS combination elemental map and SAD patterns of (**d**) $Al_{3.2}Fe$ and (**e**) $Al_5Fe_2$.

Adjacent to the steel hook at the retreating side, the interfacial zone was also examined using TEM analysis (Figures 14 and 15). As seen in Figure 14a for specimen A, the sampling position is very close to the Zn ascent. However, from the EDS elemental map (Figure 14c), no Zn was found in the interfacial zone. Several interfacial positions (i.e., Al matrix, IMCs layer, and steel matrix) for EDS analysis were also investigated. The maximum Zn content was found to be 1.06% at the IMCs layer where is near the Al-rich side. This means, with a high revolutionary pitch of 1.0 mm/rev, the low welding heat input does not cause significant dissolution of Zn in Al. Also, as seen in Figure 14b,c, the interface shows a flat microstructure; the EDS combination elemental map found the thickness of the IMCs layer to be approximately 600–700 nm. The general interfacial microstructure and thickness are similar to the interface at the stir zone (see Figure 12), indicating the similarity of the heat history at the two positions. Two kinds of IMCs were identified by SAD: $Al_6Fe$ at the Al-rich side and nanocrystalline, which is close to $Al_{3.2}Fe$, at the Fe-rich side.

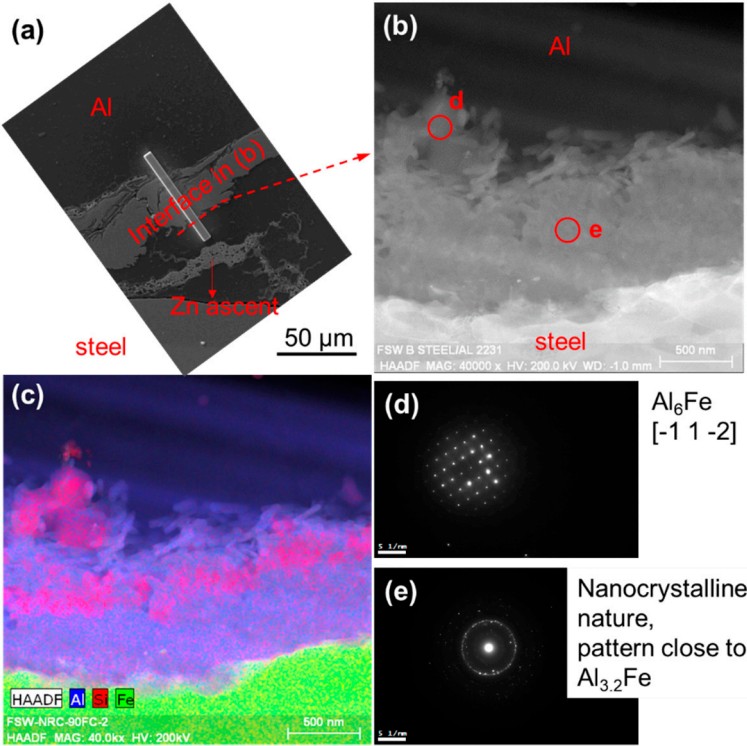

**Figure 14.** TEM micrographs of the interfacial microstructure at the hook zone of sample A: (**a**) focused ion beam (FIB) sampling position, (**b**) HAADF image, (**c**) EDS combination elemental map and SAD patterns of (**d**) Al6Fe and (**e**) nanocrystalline close to $Al_{3.2}$Fe.

The hook zone TEM images of specimen B are shown in Figure 15. From the sampling position as shown in Figure 15a, the Zn layer ascended under the impact of stirring pin at the steel hook zone. Figure 15b shows the TEM overview of the FIB specimen, the red rectangle indicates the EDS analysis location. EDS analyses of different interfacial regions in Figure 15c are presented in Table 3. Only a very small amount of Zn (0.2%) was found at location 1, i.e., for RP of 0.5 mm/rev, the dissolution of Zn is trivial or not in the Al matrix. Zn appeared at the interface from the EDS combination elemental map as shown in Figure 15d. Within the IMC layer, location 4 contains 38.3% Zn, which is much higher than the Al matrix. These results indicate that interfacial Zn is not caused by the dissolution of Zn. It could be due to the residue of Zn-coating when it was ascending with the pin rotating. This phenomenon does not occur in specimen A which is potentially due to the lower welding heat input of sample A. Note that the mean thickness of the IMCs layer is about 500 nm, which is thinner than the interface at the stir zone (see Figure 13) of sample B. It was reported that Zn promoted solid solubility of Fe in Al, which probably contributed to decreasing the intermetallic compounds in the layer structures [23]. In this work, the solubility of Fe in the Al matrix does not significantly increase, as seen in the EDS results (position 1). However, it is interesting to note that Si (~6%) and Mn (~1%) are found to diffuse from the Fe-side to the Al-side which causes aggregation in the interface (Figure 15e,g). The SAD pattern of the interface is shown in Figure 15h, but the interfacial IMC could not be identified based on the current database. From EDS analyses (location 2 and 3), it could be speculated that the IMC is Al-Fe-Si-based with ~6% Si. It was reported that, in Al/steel brazing joints, Si could suppress the diffusion of Fe from the steel-side into the weld, thus reducing the thickness of the IMC layer [25,26] and even generating Al-Fe-Si-based IMC [26]. This means the abnormal thin IMC layer could be related to the aggregation of Si and Mn and the generation of Al-Fe-Si-based IMC in the interface.

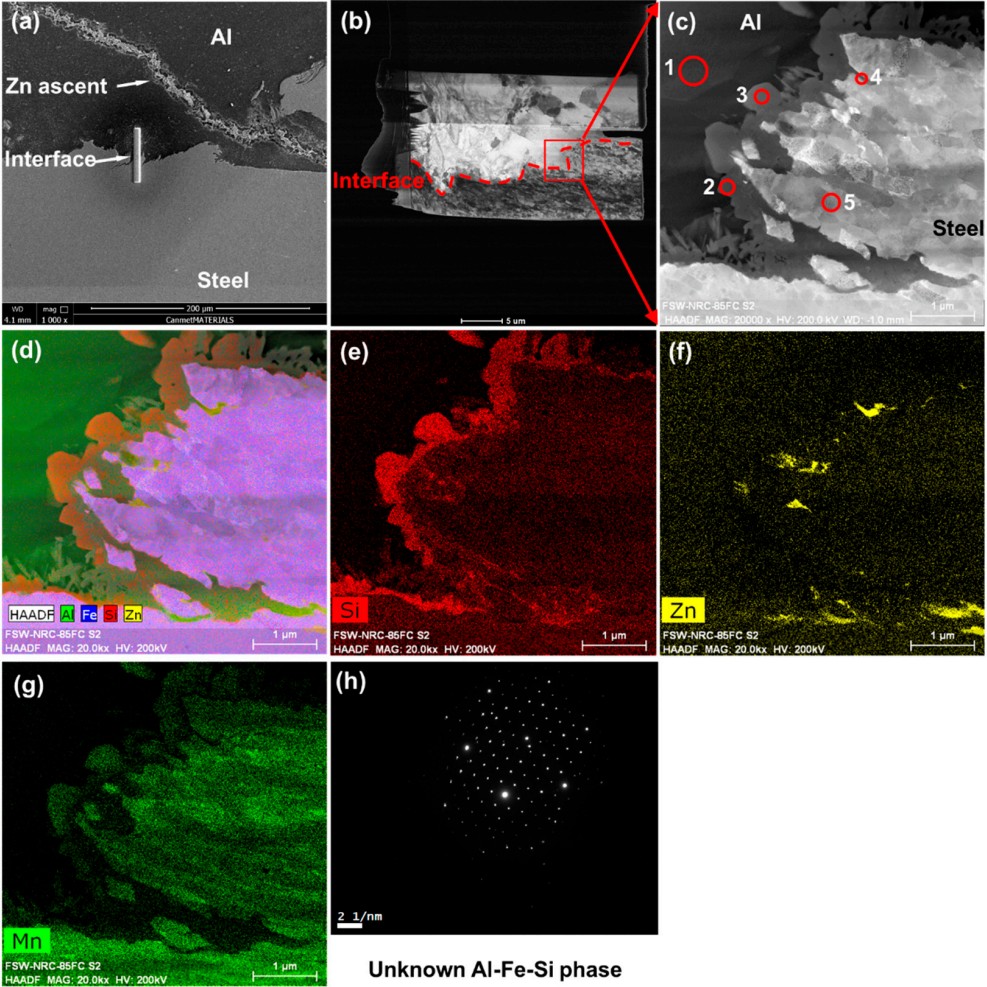

**Figure 15.** Interfacial microstructure at the hook zone of sample B: (**a**) FIB sampling position, (**b**) TEM overview of the interfacial zone, (**c**) HAADF image, (**d**) EDS combination elemental map, individual elemental maps of (**e**) Si, (**f**) Zn, and (**g**) Mn, and (**h**) SAD pattern.

**Table 3.** EDS spot analyses of the interfacial zone (see Figure 15c).

| Position | Element (at. %) | | | | | | | |
|---|---|---|---|---|---|---|---|---|
| | Al | Fe | Si | Zn | Mn | Cr | Cu | Mg |
| 1 | 98.2 | 0.1 | 0.2 | 0.2 | - | - | 0.9 | 0.4 |
| 2 | 74.0 | 17.4 | 5.9 | 0.1 | 1.1 | 0.1 | 1.5 | - |
| 3 | 75.4 | 15.8 | 5.6 | 0.1 | 1.5 | 0.1 | 1.4 | - |
| 4 | 18.0 | 21.8 | 0.6 | 38.3 | 0.6 | - | 3.5 | 17.3 |
| 5 | 0.6 | 93.5 | 0.7 | - | 1.5 | 0.2 | 3.5 | - |

## 3.3. Mechanical Properties

Owing to the fact that the steel is thicker and of much higher strength than aluminum, sound lap joints are unlikely to fracture at the steel sheet. We concentrated on the microhardness distribution of the aluminum sections and results are presented in Figure 16. In general, when friction stir welding age-hardening aluminum alloys, due to the welding heat input, the precipitates are dissolved or coarsened in the weld nugget and HAZ, these regions would be softer than the base metal. As seen from the hardness profiles, similar results have been observed. The softest region in the aluminum sheet is the junction between HAZ and TMAZ. The above results are in good agreement with the reported study of AA6082 FSWs [27].

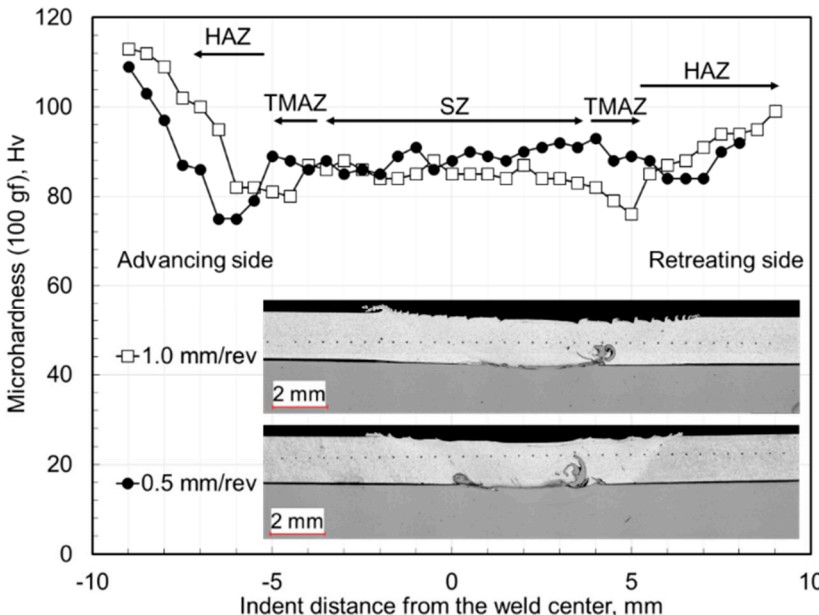

**Figure 16.** Microhardness profiles of the aluminum sections with different RPs.

However, it is interesting to note the similar microhardness values of the stir zone at 1.0 and 0.5 mm/rev. To validate these results, a tensile test was carried out and the results are presented in Figure 17. Although the hardness of the stir zone is significantly lower than that of the base metal, the stir zone shows good mechanical properties. The ultimate tensile strength (UTS) of the stir zone did not decrease significantly and the elongations were better than that of the base metal. It should be noted that the strength efficiency of the stir zone is higher than the reported result of AA6082-T6 friction stir welded butt joints [27]. In that report, the stir zone at RP of 0.25 mm/rev performed a strength efficiency of 77%. As a comparison, in this work, RPs of 0.5 mm/rev and 1.0 mm/rev resulted in strength efficiencies of 87% and 89%, respectively.

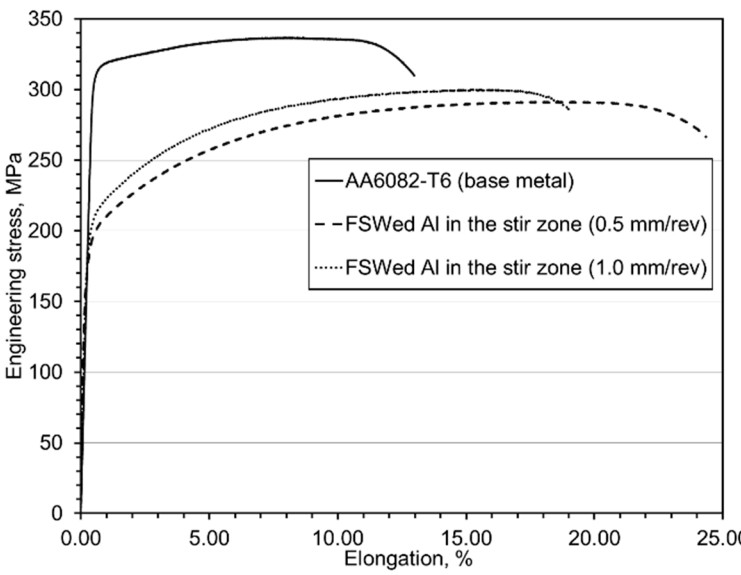

**Figure 17.** Engineering stress–strain curves of the aluminum in the stir zone compared with the base material.

For joints at 1.0 and 0.5 mm/rev, given the strong mechanical properties of SZ and the similar hardness profile of the welded Al sheet, the two joints are expected to exhibit similarly good lap shear

performance considering the interfacial IMCs layers are thinner than several micrometers and possibly enhanced the joints. However, as mentioned in Figure 6, the experimental result is different from the above guess. Figure 18 presents the lap shear curves of the two welding conditions. As can be seen, the maximum lap shear loads of three joints at 1.0 mm/rev are at a stable level. In contrast, the results of the three joints at 0.5 mm/rev are fluctuant. IMC plays a key role on the mechanical properties of the joints [28]. The thickness of the brittle IMC usually should be limited to several micrometers or even thinner to obtain acceptable joint strength. As seen from Figures 12 and 14, the revolutionary pitch determines the IMC thickness at the interfaces. In contrast, we also found that the welding parameter changes have little effect on the mechanical performance of the aluminum in the stir zone (from Figures 16 and 17). It should be noted that no interfacial fracture occurred in the specimens of 1.0 mm/rev, but interfacial fracture occurred in the specimens of 0.5 mm/rev. Therefore, we could conclude that revolutionary pitch determines the joint fracture location and lap shear specimen performance by determining the interfacial microstructure.

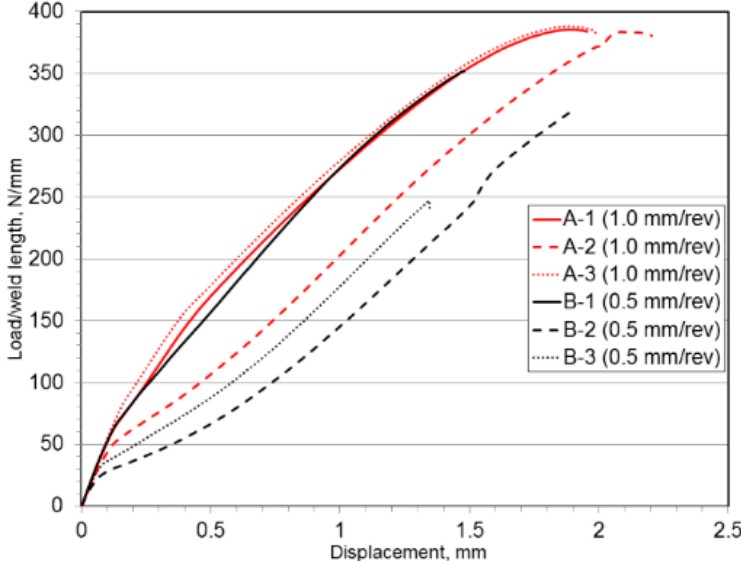

**Figure 18.** Lap shear curves at the different revolutionary pitch, joint strength expressed as the maximum failure load per millimeter of the weld.

### 3.4. Estimation of Stir Zone Mechanical Behavior

With the development of FSW and the goal of understanding the joints, researchers developed several models to predict the joint's mechanical properties such as hardness [29], UTS [30–32], elongation [32], and fatigue lifetime [33]. So far, the prediction of mechanical behavior is rarely reported. In this part, based on the experimental data of stir zone aluminum and steel, we tried to model the stir zone mechanical behavior in different aluminum/steel thickness combinations. According to the modeling results, the role of interface microstructure on mechanical properties of the FSWs was demonstrated.

The most commonly used mixture law, the iso-strain assumption, is used for modeling [34]. In this assumption, the strain of each constituent is taken equally. Correspondingly, in the stir zone, the constituents could be divided into three kinds: welded Al, welded steel, and interfacial IMCs. For easily running the modeling, we assume that no phase transition during the deformation and ignore the slight content of Zn in the interface. Thus, the stress equation could be given by

$$\sigma(\varepsilon) = V_{Fe} \cdot \sigma_{Fe}(\varepsilon) + V_{Al} \cdot \sigma_{Al}(\varepsilon) + (1 - V_{Fe} - V_{Al}) \cdot \sigma_{IMC}(\varepsilon) \tag{1}$$

where $\sigma_{Fe}(\varepsilon)$, $\sigma_{Al}(\varepsilon)$, and $\sigma_{IMC}(\varepsilon)$ are the flow stresses of steel, aluminum, and IMC, respectively, $V_{Fe}$ and $V_{Al}$ are the volume fractions of steel and aluminum, respectively.

It should be highlighted that different kinds of IMCs have different mechanical responses so it is difficult to quantify those in practice. In order to continue to simplify the modeling, we could treat the microscale IMCs layer as the strong boundary between Al constituent and steel constituent. Thus, the equation could be given as

$$\sigma(\varepsilon) = V_{\text{Fe}} \cdot \sigma_{\text{Fe}}(\varepsilon) + (1 - V_{\text{Fe}}) \cdot \sigma_{\text{Al}}(\varepsilon) \tag{2}$$

By conventional tensile and modified shear testing, the stress–strain curves of basic constituents were obtained and presented in Figure 19. It should be noted that the true strains of tensile data are smaller than those of the shear test. For comparison, by calculating the reduction of area at fracture, the stress–strain curves of tensile data were thus lengthened. Results clearly show that the aluminum stresses of tensile data are higher than those of shear-test data. The mechanical behavior of welded steel is quite dependent on the weld parameters. The experimental data of aluminum in the stir zone was used to model the mechanical response of friction stir welded aluminum. The conventional tensile experimental data of steel in the stir zone was used to model the mechanical response of friction stir welded steel.

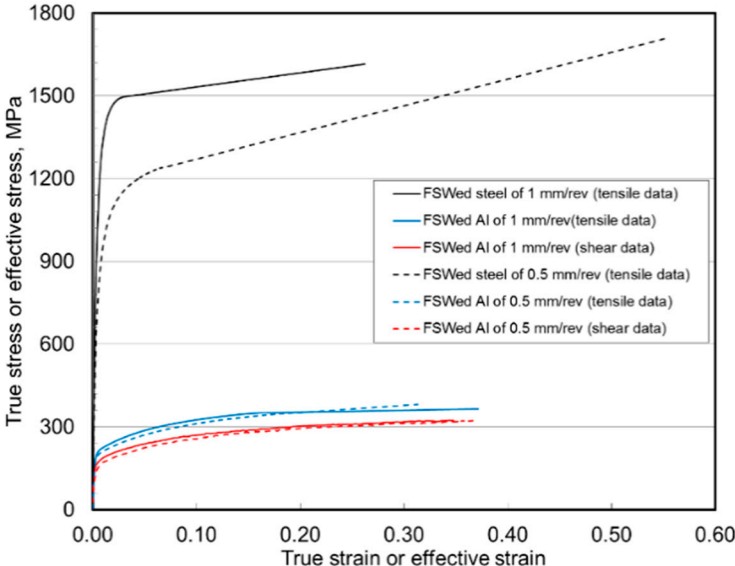

**Figure 19.** True and effective stress–strain curves of constituents in the stir zone obtained from tensile and shear test.

For tests on specimens with different RPs, the modeling results are presented in Figure 20 along with the experimental ones. It should be noted that no delamination occurred during shear testing owing to the mentioned microscale interfacial IMCs layer. As we know, FSW process is a solid-state welding process with intense mixing and stirring. Thus, the rotating pin has the potential to introduce some small steel fragments from the steel matrix into the aluminum matrix. Note that the volume fractions of the constituents are measured by OM as previously described. The local volume fractions of two constituents may vary within the joint. In addition, we ignored the mechanical responses of Zn and IMCs. In consideration of these factors, the modeling results are in good agreement with the experimental ones indicating the resultant interfaces act as good as grain boundaries. This part work prospects an easy way to estimate the joint mechanical response using a modified shear test.

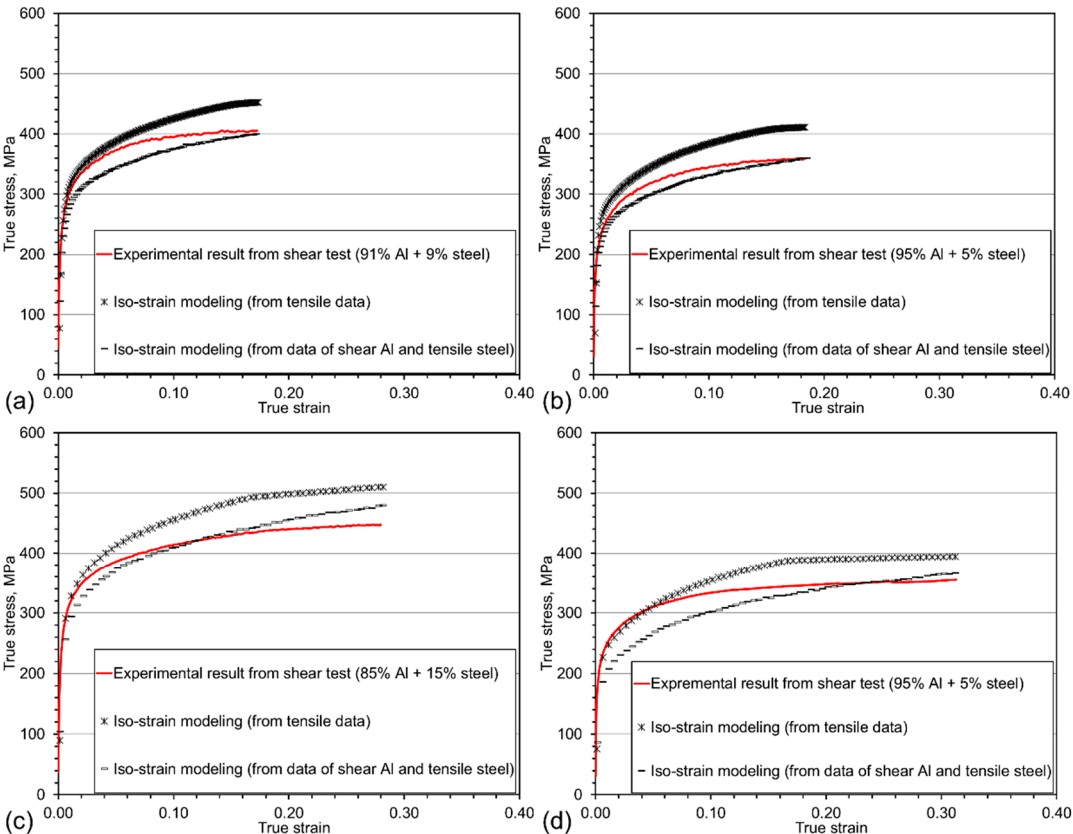

**Figure 20.** Linear modeling compared with the experimental results at different RPs. (**a**,**b**) at RP of 1.0 mm/rev with different volume fractions and (**c**,**d**) at RP of 0.5 mm/rev with different volume fractions.

## 4. Conclusions

The dissimilar lap joining of 1.5-mm thick AA6082-T6 sheet to 2.0-mm thick galvanized Zn-coated DP800 sheet was successfully conducted using friction stir welding process. The interfacial IMCs and their effect on the mechanical properties of FSWs were evaluated. The conclusions can be drawn as follows.

- It is feasible to lap join the 6082-T6 aluminum alloy to DP800 dual-phase steel sheets. Under the welding conditions in the present study, maximum joint strength reaches 71% of that of the base Al material with a RP value of 1 mm/rev (1250 RPM and 1250 mm/min).
- In the stir zone, $Al_{3.2}Fe$ in the Al-rich side and $Al_5Fe_2$ in the Fe-rich side were detected for two welding conditions studied. In the hook zone, however, $Al_6Fe$ was detected in the Al-rich side and nanocrystalline pattern close to $Al_{3.2}Fe$ at RP of 1.0 mm/rev. Under a relatively low RP (0.5 mm/rev) in the hook-zone, zinc was found at the interface with the aggregation of Si and Mn elements at the Al-rich side of the interface which leads to the generation of Al-Fe-Si phase thus decreases the thickness of IMCs layer.
- In the stir zone, RP has a significant influence on the interfacial microstructures. The interfacial IMCs layer at an RP of 1.0 mm/rev is simple and flat but the one at RP of 0.5 mm/rev becomes thicker and more complex where IMCs are scattered in the Al matrix. However, the stir zone aluminum, under different RP values, is similar in microhardness value and tensile behavior.
- The iso-strain-based linear mixture law was used to model the stir zone mechanical response. The modeling results are in good agreement with the experimental ones, indicating the microscale IMCs act good as strong boundaries of dissimilar materials.

**Author Contributions:** Funding acquisition, J.K., F.N., and Y.C.; Investigation, J.K., S.L., B.S.A., and F.N.; Methodology, J.K., S.L., and F.N.; Supervision, J.K. and Y.C.; Writing—original draft, S.L.; Writing—review & editing, J.K., Y.C., and F.N.

**Funding:** The authors acknowledged the financial support from Canadian Federal Government Interdepartmental Program on Energy R&D (PERD) and CanmetMATERIALS, Natural Resources Canada. Yuhua Chen and Shuhan Li also express thanks for the financial support from the National Natural Science Foundation of China, grant number 51865035; Fund for Jiangxi Distinguished Young Scholars (2018ACB21016), the Project for Jiangxi Advantageous Scientific and Technological Innovation Team (20171BCB24007 & 20181BCB19002); and Aviation Science Funds of China, grant number 2017ZE56010.

**Conflicts of Interest:** The authors declare no conflicts of interest.

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
