# Peer review of "Effect of Revolutionary Pitch on Interface Microstructure and Mechanical Behavior of Friction Stir Lap Welds of AA6082-T6 to Galvanized DP800"

_metals, doi:10.3390/met8110925_

Round 1
Reviewer 1 Report
The authors have presented an interesting paper on Friction Stir Welding of two different materials, namely aluminum and steel, and have studied the effect of revolutionary pitch applied in the process on the microstructure of the interface of the two materials. Furthermore, they have studied the mechanical properties of the specimens that they produced. The paper is useful and interesting due to the potential application in industrial practice. However, some revisions are required before the paper can be accepted for publication.
First of all, some pats of the manuscript require language revision. It is advised that the authors proofread their paper and make appropriate changes.
At the introduction section the authors need to describe what are the works presented in this area with focus on the problems and why the work performed is needed. This way, the authors will explain and clearly state what is the novelty of the work they propose.
The authors need to elaborate more on the selected friction stir welding conditions. How and why were these conditions selected? Please, elaborate more. Furthermore, please correct "electron discharge machining" with either "electro" or "electrical" discharge machining. Furthermore, wire electrical discharge machining is usually abbreviated as WEDM.
In section 3, the authors mention "in accordance with ...2.25 kN was determined". Please explain.
In the text between Fig. 5 and Fig. 6, some results are described but no explanation is provided. Are there any references or theories to support the statements? On what grounds were the conditions in Table 2 selected?
The results reported is section 3 are for 2 case studies. The results are few and in some tests the conclusions are incomplete e.g. Fig. 17. The authors should comment on that.
Finally, the results of Fig. 20 show discrepancies between the results in several parts of the graph. Please, be more specific on the comments of this graph.
Author Response
Thanks for the comments from the reviewer for the improvement of the manuscript.
Point by point responses could be found in the attachment.

Reviewer 2 Report
Introduction. Although “IMC” has been defined in the Abstract, it should be defined again when first appearing in the main text. This section provides excellent background for the study; no substantive changes are recommended other than polishing the English presentation.
Experimental Procedures. (1) The sources of the aluminum and steel alloys should be indicated. (2) The manufacturer and model for the friction stir welding machine should be provided. (3) Were the different travel speed and rotation speed conditions employed for the friction stir welding experiments determined from preliminary experiments, or was guidance provided by the FSW machine manufacturer? (4) The manufacturer and model information should be included for the optical microscope, scanning electron microscope, and transmission electron microscope. Brief statements about how the TEM specimens were prepared should be included. (5) It is important to indicate the number of replicate test specimens for the different characterization and property evaluation experiments.
Results and Discussion. (1) For the information presented in the highly important Figures 5 and 6, there should be comments about how many test specimens were employed for each of the data points, which presumable represent mean values, and about the variability among replicate test specimens for each condition. (2) In Figure 8(e), are standard deviations being presented along with the bars for mean values? The number of replicate test specimens (N) should be indicated. (3) Does the accuracy of the EDS spot analyses warrant reporting the elemental concentrations in Table 3 to two decimal places? (4) The terms in the modeling Equations (1) and (2) should be defined. (5) A closing paragraph is recommended, in which the major limitations of the study are summarized and suggestions for future research are provided.
Author Response

(The authors gave the same response as above.)

Reviewer 3 Report
Dear auhors,
This has been a great pleasure to review your paper extremely well written with a real effort to provide clear explanations.
Regards

Author Response

(The authors gave the same response as above.)
